# Exploring determinants of hydrocele surgery coverage related to Lymphatic Filariasis in Nepal: An implementation research study

**Choden Lama Yonzon**[1]*, **Retna Siwi Padmawati**[1], **Raj Kumar Subedi**[2], **Sagun Paudel**[1], **Ashmita Ghimire**[1], **Elsa Herdiana Murhandarwati**[3]

1 Department of Health Behavior, Environment, and Social Medicine and Centre of Health Behavior and Promotion, Faculty of Medicine, Public Health, and Nursing, Universitas Gadjah Mada, Yogyakarta, Indonesia, 2 Bhaskar-Tejshree Memorial Foundation, Kathmandu, Nepal, 3 Postgraduate Program of Tropical Medicine, Faculty of Medicine, Public Health, and Nursing, Universitas Gadjah Mada, Yogyakarta, Indonesia

* yonzon.chhoden.cy@gmail.com

**Data Availability Statement:** All relevant data (De-identified and categorised) are within the paper and its Supporting Information files. Full and raw

## Abstract

### Background

Hydrocele is a chronic condition in males in which there is an excessive collection of straw-colored fluid, which leads to enlargement of the scrotum. It is a common manifestation of lymphatic filariasis (LF) affecting nearly 25 million men worldwide. Surgery is the recommended treatment for hydrocele and is available free of cost in all government hospitals in Nepal. This research explored patient, provider, and community factors related to accessing hydrocele surgery services by the patients.

### Methods

This study employed a qualitative method. The research was conducted in two LF endemic districts, namely Kanchanpur and Dhading, which are reported to have the highest number of hydrocele cases during morbidity mapping conducted in 2016. In addition to five key informant interviews with the LF focal persons (one national and 4 district-level), nine in-depth interviews were conducted with hydrocele patients (5 of whom had undergone surgery and 4 who had not undergone surgery) and with 3 family members, and two focus group discussions with the female community health volunteers.

### Results

Most of the respondents did not have knowledge of hydrocele as one of the clinical manifestations of LF nor that it is transmitted through a mosquito bite. Although perceived as treatable with surgery, most of the patients interviewed believed in as well as practiced home remedies. Meanwhile, fear of surgery, embarrassment, lack of money, along with no knowledge of the free hydrocele surgery acted as barriers for accessing the surgery. On the other hand, financial support, flexible guidelines enabling the hospital to conduct surgery,

transcripts cannot be shared publicly due to confidentiality issues involving personal details of human subject. In order to request access to the data, please contact Emilia Sri Wulandari, emilia@ugm.ac.id, or Yuyun Yohana, yuyun.yohana@ugm.ac.id.

**Funding:** This work was funded provided by, the Special Programme for Research and Training in Tropical Diseases based at the World Health Organization in Geneva (WHO-TDR), Switzerland. The corresponding author (CLY) received specific funding for this work from WHO/TDR special program for implementation research. The funders had no role in the study design, data collection and analysis, decision to publish, or preparation of the manuscript.

**Competing interests:** The authors have declared that no competing interests exist.

decentralization and scaling up of morbidity mapping along with free hydrocele surgery camps in any remaining endemic districts were identified as enablers for accessing surgery.

## Conclusion

Hydrocele surgery coverage could be improved if the program further addresses community awareness. There is a need for more focus on information dissemination about hydrocele and hydrocele surgery.

## Introduction

Lymphatic filariasis (LF) is a mosquito-borne, highly disfiguring parasitic disease and is considered as one of the major public health problems in 73 countries worldwide, including Nepal [1]. One-third of the people affected with the disease live in India, one-third in Africa and most of the remainder are in South Asia, the Pacific and the Americas [2]. Filarial infection can damage patients' lymphatic system causing pain, known as acute dermatolymphangioadenitis (ADLA) due to secondary infection of lymphoedematous tissues, chronic disfiguring and disabling conditions including hydrocele (scrotal swelling), lymphoedema (tissue swelling) and elephantiasis (skin/tissue thickening) of limbs [3]. In 2000, about 40 million people were disfigured and incapacitated by the disease, of which, there were 25 million men with hydrocele and 15 million people with lymphoedema. LF is a neglected tropical disease (NTD) and is considered to be one of the most common causes of long-term disability [4].

In line with the Global Program to Eliminate Lymphatic Filariasis (GPELF) launched by the World Health Organization (WHO) in 2000, Mass Drug Administration (MDA) and Morbidity Management and Disability Prevention (MMDP) are the two main strategies adopted by the government of Nepal to eliminate LF as a public health problem by 2020 [5]. MDA involves an annual provision of a combined dose of medications (DEC and Albendazole) to all eligible persons living in endemic areas for at least five years and MMDP involves a basic package of recommended health services which includes treating ADLA, surgery for hydrocele to prevent progression of lymphoedema to ADLA. As per the WHO guidelines, for endemic countries to successfully initiate a morbidity mapping program, morbidity data should be collected at least annually and include information relating to the estimated number of patients who have lymphoedema, hydrocele and ADLA, in addition to the actual number of those treated for these manifestations. For the GPELF to succeed in eliminating LF as a public health problem, achieving 100% geographical coverage of both MDA and MMDP is necessary [2].

Hydrocele is a chronic condition in men in which there is an excessive collection of straw-colored fluid in the tunica vaginalis, a two-layer sac that holds the testes and epididymis and the scrotum enlarges to various sizes, in rare cases obliterating the entire penis [2]. An increase in age prevalence is seen in hydrocele cases, as reported in most Asian and African sites, with as high as 50% prevalence seen in older age groups (above 45 years) and the size of hydrocele increases with age [6]. As much as the physical disability, the condition is also associated with significant social stigma, impact on marriageability, men's physical and sexual function, and lower employment opportunity resulting in lowered economic input in household activity and family discord [6, 7].

As per the GPELF goal of 2020, many countries have scaled-up surveillance and morbidity management activities to satisfy WHO LF elimination dossier components required for validation [8]. Accordingly, Nepal also laid out and implemented both MDA and MMDP

interventions; with nearly 82% (50 out of 61 districts) of endemic districts having stopped MDA and 14 districts completed Transmission Assessment Survey III (TAS III) [9]. Meanwhile for the MMDP component, it is also gradually scaling-up with morbidity mapping planned to cover all the endemic districts by 2020, along with hydrocele surgery coverage. As per the WHO guidelines, providing hydrocele surgery is the minimum recommended service and care for the hydrocele cases. While MMDP services such as hydrocele surgery, symptomatic treatment, management of acute attacks as well as home-based self-care instructions provided by the Female Community Health Volunteers (FCHVs) have been available since the beginning of the LF Elimination program [1], there were challenges as these services were provided as a mainstream health care services in the government health facilities. It often meant that cases like hydrocele surgery, could not be performed due to lack of skilled doctors and infrastructure in some of the district hospitals. Hence, it is only recently that the Ministry of Health and Population had scaled-up the MMDP component of Nepal's LF Elimination Program by conducting active mapping of the LF morbidity cases in the endemic districts beginning in 2016 and focusing on hydrocele cases exclusively by conducting free hydrocele surgery in a camp-style approach in respective district hospitals with a separate budget allocation [10]. This camp-style approach was also recommended by the WHO per one guideline published in 2002 [6]. With this method, the patients are informed and referred to the free surgery camp through communication channels such as pamphlets, radio messages as well as through FCHVs prior to the camp.

In Nepal, LF baseline mapping conducted between 2001–2002 had reported approximately 20,000 hydrocele cases of LF. Morbidity mapping conducted in 2016 in only 12 (out of 61 endemic) districts has identified nearly 9,000 cases of hydrocele. Additionally, the latest update from the LF elimination program also states that 7,327 hydrocele surgeries have been performed across the country till 2018. Another challenge is the data of the number of surgeries conducted per district relative to the number of actual cases are still not available [1]. Based on the cases identified from just 12 endemic districts, it can be predicted that there are still many cases that need surgery and attention despite being available for free with expanded services.

Many effective and proven interventions fail to translate into meaningful patient outcomes across multiple contexts [11]. A WHO report of barriers and enablers of effective coverage from the country of Moldova states that, for each case that is not detected or treated, there are individual, community and health system factors that have contributed to the existing barriers to healthcare [12]. One recent study in Sri Lanka, where LF has already been eliminated as a public health problem, identified major operation challenges in implementing the MMDP component post-LF elimination phase such as lack of coverage of the services in the endemic regions, personnel shortages, especially staff with significant knowledge and expertise, distance of health facility from the community, and information dissemination [13]. Similar findings have been reported from one study in India, which identified the lack of advertisement as an important missing piece in the morbidity management program and recommended the use of information, education and communication (IEC) materials, especially targeting the populations from poor and less educated backgrounds [14]. One study in Nepal found little to no information on insights of healthcare seeking behavior, access to care, and self-care practice of LF patients. The active case finding, referral, and treatment of the LF patients are further complicated by the lack of integrated reporting of private hospitals [15]. This study aimed at understanding the barriers and enablers to accessing hydrocele surgery to facilitate and expedite the national LF elimination goal with necessary policy recommendations and thereby integrating hydrocele patients into society free from disease and disability.

## Materials and methods

### Research type and design

The study employed an exploratory effort to understand the knowledge and perception of hydrocele and barriers and enablers of hydrocele surgery among the patients, and care providers. A qualitative study design was used by adopting the ecological framework developed by Durlak and Dupre in 2008 [16]. Consolidated criteria for reporting qualitative research (COREQ) checklist was used to report the methods used in this study (S1 Table). Details of data collection methods are as follows:

**a. Key Informant Interview (KII).** KIIs were conducted with the stakeholders/focal persons of the LF elimination program from the central level in Kathmandu, district health offices and district hospitals in Kanchanpur and Dhading districts. Stakeholders are the focal persons of the LF elimination program in Nepal with decision-making authority.

**b. Focus Group Discussion (FGD).** FGDs were conducted with the FCHVs in both Kanchanpur and Dhading districts.

**c. In-Depth Interview (IDI).** IDIs were conducted with hydrocele patients (both with and without surgery) and family members (any immediate close family member/wife) of the patients who had not undergone surgery.

### Research setting and time

The selected districts lie in two of the seven provinces of Nepal. The study was conducted in Kanchanpur and Dhading Districts of Nepal which are classified as endemic for LF. Kanchanpur District with an area of 1,610 square kilometres, has a total population of 451,248 as of the 2011 census and lies in Province No. 7 in the far-western region. It is bordered by another two districts on the east and north, and with India on the south and west border. Dhading is located in the hilly region of Province No. 3 of the central region of Nepal, and covers an area of 1,926 square kilometres with a population of 336,067 as per the 2011 census [17]. Kanchanpur and Dhading have the highest number of hydrocele cases based on the latest morbidity mapping survey, out of 12 endemic districts mapped till 2016 [1]. So far, three and two free hydrocele surgery camps had been conducted in Kanchanpur and Dhading districts, respectively by the time of data collection. The research was conducted between June-August 2019.

### Sampling and sample size

**Key Informant Interviews (KIIs).** Five KIIs were conducted with the stakeholders of the LF elimination program. One stakeholder from the central level and 4 stakeholders (2 from each district) from the district health office and district hospital in Kanchanpur and Dhading were selected.

**Focus Group Discussions (FGDs).** Two FGDs were conducted, one in each district. FCHVs were identified in consultations with the district health offices and were contacted and selected based on their availability during the time of data collection. Seven FCHVs in Kanchanpur and 5 FCHVs in Dhading participated in the FGDs.

**In-Depth Interviews (IDIs).** Both purposive and snowball samplings were used for selecting IDI respondents. With the suggestions of district stakeholders, villages were purposefully selected based on the number of cases and their proximity to the district headquarters. With the help of FCHVs as well as the registry from the district health offices and district hospitals, patients with hydrocele were located and approached for interviews. In addition, a few hydrocele patients were also identified with the help of hydrocele patients who participated in the interviews. Twelve IDIs in total from both districts were conducted which included nine

hydrocele patients (5 of whom has had surgery and 4 of whom did not have surgery at the time of the interview) and three family members of hydrocele patients. Family members of hydrocele patients who had not yet undergone surgery were included in order to better understand family as well as community perspectives on the possible barriers to seeking care. Hydrocele patients who were recent migrants (less than 6 months) or temporary residents to the area, below 18 years of age and who had undergone surgery less than 6 months prior were not included in the study. The six-month timeframe was chosen with the assumption that the patient will have fully recuperated after the surgery. Telephone inquiries were done with the identified and potential respondents prior to the interviews to obtain their consent and time availability. Respondents were not known to the interviewer prior to interviewing. Informed consents were obtained from all the participants.

## Data collection and research instruments

Stakeholders from the central level were consulted starting from the inception of the study. With their suggestions, stakeholders in both districts were approached for data collection. Focus group discussion, KII and IDI guidelines were developed in order to better address the research questions through the identified variables and to more completely present the findings in thematic order. The KII, IDI and FGD guidelines were developed based on the available literature of studies related to LF and hydrocele and were aligned with the objectives of the study. These guides were first developed in English and then translated into the Nepali language and reviewed for linguistic reliability and correctness in consultation with a local supervisor (RKS). Data triangulation was done for maintaining the validity of the tools by cross-checking data from the different group of respondents: IDI, KII and FGD. All of the interviews and FGDs were conducted with the help of a voice recorder. On an average, IDIs and KIIs lasted about 30–45 minutes and the FGDs took about one and half-hour.

All IDIs and KIIs were conducted in the respondent's home and/or office in the local language. IDIs with hydrocele patient were conducted by a male research assistant who had nearly a decade of experience with data collection methods and is a university graduate from Nepal, (due to the sensitivity and hesitancy by male respondents towards the female lead author, CLY), with the help of the corresponding author (CLY). The research assistant was oriented about the study, objectives, methods, data collection tools, data management, interview techniques, and ethical issues prior to mobilization in the field. The corresponding author (CLY-student at Universitas Gadjah Mada) conducted the KIIs and FGDs. Data were collected until saturation, after making sure that all the questions and variables were covered from all groups of respondents and no new information was gathered. No observers were present at the time of the interviews. Debriefing was done at the end of each interview.

## Data analysis

Interviews were recorded only after getting verbal and written consents from the participants. In addition, field notes were also taken to clarify and confirm responses. The data were transcribed verbatim within 24 hours by the principal interviewer (IDIs with hydrocele patients were transcribed by the research assistant as he carried out the interviews) in order to maintain clarity and avoid losing and missing any information. Transcriptions were further cross-checked with the field notes if and when necessary to ensure data quality and completeness. Transcribed data were further checked, then re-checked to ensure data quality by CLY, by going completely through the data recordings and transcriptions. Transcribed data were then translated into English and then read, and re-read to identify codes and themes as per the objectives and variables of the study. The data were then coded and grouped into various

categories and sub-categories or themes. Thematic analysis of the data according to the research objectives was done by identifying similar patterns in responses. The corresponding author (CLY) did all the data coding and analysis manually. RSP and EHM helped to oversee the data processing.

### Research ethics

Ethical approval was given by the Ethical Review Board of the Nepal Health Research Council in Nepal on May 16[th], 2019 and the Medical and Health Research Ethics Committee of Universitas Gadjah Mada, Yogyakarta, Indonesia on July 23[rd], 2019.

## Results

### Sociodemographic characteristics of the respondents

All of the five stakeholders interviewed (KII) were males who had more than two years of experience working in the same position (as a focal person in the LF elimination program) except one who had just 1.5 years of experience. Talking about hydrocele patients, both with and without surgery, the study encountered patients predominantly in a higher age group, with age range of 40–74 years and median age of 57 years. A total of 15 hydrocele patients were approached of which 9 agreed to participate in the study. Among the participants who had undergone surgery, 1 had surgery done through a hydrocele camp in Kanchanpur, 2 at private hospitals in Kathmandu and the remaining 2 in India. More details of respondents' sociodemographic information are provided below in Table 1.

### Knowledge and perception of hydrocele (IDI and FGD)

Almost all of the hydrocele patients and their family members and even some FCHVs did not know the cause of hydrocele was resulting from lymphatic filariasis and that it is transmitted by the bite of a mosquito. One respondent in Kanchanpur, who also happened to be an employee at the district health office, knew about its cause and mode of transmission. In Dhading, the patients attributed hydrocele to cold temperature, and thus the patients avoided getting cold or going out in the rain. In Kanchanpur, some respondents attributed the cause of hydrocele to heavy physical strenuous works such as riding bicycle, rickshaw, while others attributed it to accidental trauma, and injury to the scrotum.

*". . .. I used to ride cycle a lot. I [initially] thought maybe it [hydrocele] was because I used to ride too much cycle. Later I found out many cases are caused by the bite of a mosquito."* (IDI post-surgery, Kanchanpur)

*"To be honest, I still don't know the cause of hydrocele. They say it is because of the cold, that is the common belief around here."* (FGD, FCHV, Dhading)

Most of the respondents have developed hydrocele dating back as far as one year to 20 years, and it progressed and increased in size with age. They mentioned having difficulty in doing mundane day to day activities such as simple walking, bathing and working. Because the severity and size of hydrocele tend to grow with time, the degree of challenges seemed to vary among the respondents. Due to the pain and obvious visibility, they had clothing restrictions, as quoted in the following responses:

*"It is difficult while bathing and going to some religious ceremonies and interacting with friends. We have to wear a dhoti [sarong]. When I wear dhoti with only underpants, scrotum*

**Table 1. Sociodemographic profile.**

| Characteristics | Stakeholders | | | | Hydrocele patients | | | | Female Community Health Volunteers (FCHVs) | | |
|---|---|---|---|---|---|---|---|---|---|---|---|
| | Total | Kanchanpur | Dhading | Central | Total | Pre-surgery | Post-surgery | Family members | Total | Kanchanpur | Dhading |
| **Sex** | | | | | | | | | | | |
| Male | 5 | 2 | 2 | 1 | 9 | 4 | 5 | - | - | - | - |
| Female | - | - | - | - | 3 | - | - | 3 | 12 | 7 | 5 |
| **Age (years)** | | | | | | | | | | | |
| 35–44 | 2 | 1 | 1 | - | 3 | 1 | - | 2 | 9 | 6 | 3 |
| 45–54 | 3 | 1 | 1 | 1 | 3 | - | 3 | - | 2 | 1 | 1 |
| 55–64 | - | - | - | - | 3 | 1 | 1 | 1 | 1 | - | 1 |
| >65 | - | - | - | - | 3 | 2 | 1 | - | - | - | - |
| **Education** | | | | | | | | | | | |
| Illiterate | | - | - | - | 4 | 2 | 1 | 1 | - | - | - |
| Literate | | - | - | - | 2 | - | 2 | - | 6 | 5 | 1 |
| Primary | | - | - | - | 4 | 2 | 1 | 1 | 6 | 2 | 4 |
| Secondary | | - | - | - | 2 | - | 1 | 1 | - | - | - |
| Higher secondary | 5 | 2 | 2 | 1 | - | - | - | - | - | - | - |
| **Occupation** | | - | - | - | | | | NA | | NA | NA |
| Unemployed | | | | | 1 | 1 | - | | | | |
| Farmer | | - | - | - | 2 | - | 2 | | | | |
| Laborer | | - | - | - | 1 | - | 1 | | | | |
| Driver | | - | - | - | 2 | 2 | - | | | | |
| Other | | | | | 3 | 1 | 2 | | | | |
| **Work experience of stakeholders** | | | | | NA | NA | NA | NA | | | |
| <2 years | 5 | - | 1 | - | | | | | | | |
| 3–9 years | | 1 | 1 | - | | | | | | | |
| 10–19 years | | 1 | - | 1 | | | | | 12 | 7 | 5 |

*moves around and due to friction, it hurts. I cannot wear small/tight clothing as well."* (IDI, post-surgery, Kanchanpur)

Some patients mentioned about not being able to work and loss of income due to pain and shame. Although people do not directly discriminate against hydrocele patients, they would be the topic of gossip and societal scrutiny. Because of this, patients experienced having low self-esteem, being self-conscious and being confined at home on many occasions.

*". . .I had difficulty in working as well. I couldn't work. You know in labor work, there are females in working place. People would make fun and talk things about me. Hence, I stopped working for some days. I felt very uncomfortable. Then after nearly one year of having surgery, I started working again."* (IDI, post-surgery, Kanchanpur)

*"I stopped going to my friends and my relatives. It [hydrocele] would be seen clearly if I wear trousers. I felt embarrassed in front of the sisters. So, I stopped going anywhere altogether. I had difficulty in working as well. People would make fun and talk things about me."* (IDI, post-surgery, Kanchanpur)

In most cases, there were no visible and direct discrimination nor stigmatization towards hydrocele patients. The patients themselves and even the larger community seem to know that

hydrocele can be treated, and surgery is the recommended method of treatment. However, self-stigmatization and shame were inherently attached to the persons affected, because it involved sensitive information about the genital organs of the males.

> "...There is no discrimination toward hydrocele patients. Almost everyone believes it can be treated with surgery and the swelling is due to accumulation of fluid." (FGD, FCHV, Dhading)

Maybe due to the nature of the disease, it was intriguing to find that some respondents considered that hydrocele was sometimes associated with infertility and sexually transmitted infections (STIs) as well as marriageability.

> "Some people used to say, 'you have this big scrotum, but you got married. I am not sure whether you will have children or not'. They used to suggest me to get it examined since I might not be able to have a child." (IDI, post-surgery, Kanchanpur)

FCHVs admitted that most hydrocele patients do not open up or even accept having discussion about their hydrocele. Since FCHVs are females, it might have to do with gender; a male discussing a disease in his male organs with a female is culturally sensitive, and hence some men also found it offensive when asked about it.

> "Normally, men are very ashamed about it. They are offended if we ask about it directly, and they usually deny. When we talk with their wives, then they will disclose." (FCHV, FGD, Kanchanpur)

When confronted, some would also deny having it and simply brush off the questions. It was only when immediate family and friends noticed the swelling, then the patient would talk about it and consider getting the necessary help. Otherwise, they wished to keep their private condition to themselves.

> "One of my relatives had hydrocele., but he would always deny it saying that there is nothing wrong with him and we shouldn't interfere. Later, when there was [hydrocele] camp, he went with one of his brothers [for surgery] whom we had informed about the camp." (FGD, FCHV, Kanchanpur)

A series of home remedies were also informed to have been practiced and still being practiced in both the districts but to what degree, differed from individual to individual. Most of the home remedies were practiced for easing and ameliorating the pain and swelling, and when it did not work, which was the case admitted by most of the patients, they only then sought for professional help.

> "...Like potato is a cold thing right, it contains water. So, people say that you should not consume potatoes and meat. Even I gave up eating meat. Someone told me, rice is more beneficial than chapati [flat bread]." (IDI, pre-surgery, Kanchanpur)

Most of the patients who had undergone surgery admitted that those home remedies were ineffective as they look back now, while those who do not have surgery yet, admitted to still practicing until recently. In addition, most of them took medicines at some point or once in a while for pain relief and to carry on with their lives while avoiding surgery.

*"Due to cold, scrotum swells. That's why they warm swollen part with heated bricks wrapped in cloth. They sit on top of heated bricks. We cannot say that it doesn't work because I also don't know whether it works or not, to be honest. They also avoid eating potato, and tomato."* (FGD, FCHV, Dhading)

These findings reflect the social stigma regarding hydrocele among the patients themselves which deeply affects their self-confidence level and explains the lengths they would go to hide the condition until they could find the treatment themselves, primarily at home. Respondents expressed that since the problem involves a man's genitalia, it is hard to open up and talk about it to just anyone. The LF Elimination Program in Nepal should better address the need of proper information dissemination because patients have no reliable sources to find out about their condition leaving them vulnerable to ineffective home remedies resulting in low self-esteem as well as physical restrictions.

## Knowledge and perception of free hydrocele surgery program

**Stakeholders of LF elimination program.** The focal person from the central level explained that when the program was designed to be implemented from 2016, in order to meet the dossier component of WHO for LF elimination declaration, they had targeted and expected to complete the program by 2020. When asked about how and whether the program addresses any community-level awareness activities, stakeholders had a common perception that hydrocele does not have any taboo and stigma attached to it, which was also supported and explained by the FCHVs as well. As a result of this misperception, only the institution-level program of providing free hydrocele surgery was considered enough to address the morbidity of hydrocele patients, as voiced by the stakeholders in the central level.

*". . .to be honest, we have not found and witnessed such stigma and discrimination towards hydrocele patients from the community. One reason we think is that people know it is not a communicable disease and is also not associated with mortality."* (KII, Central Level)

Regarding the possibility and necessity of providing other additional incentives to patients such as transportation cost (as a form of motivation), the stakeholders explained that hydrocele surgery is a simple surgery that does not involve admissions in the hospital and has few if any post-surgery complications. That is why the program has provisions of only free surgery.

*". . .we feel that for patients who have been living with the condition for such a long period of time, getting free treatment is in itself a big thing."* (KII, Central Level)

Focal persons from Dhading and Kanchanpur both agreed that there are very limited time and budget for information dissemination regarding upcoming hydrocele camps among the public. In addition, budget dissemination for conducting the programs is uncertain, because there is no fixed schedule of budget allocation, which leads to uncertainty in organizing the camp. When the camp was finally organized, due to limited time and budget constraints, they felt that the patient turnout rate was hampered by the lack of community awareness. The district stakeholders stressed that more budget and time is needed for advocating and advertisement of camp.

*". . .with the budget we conducted interaction workshop, printed pamphlets, run ads on radio, gave allowance for the meeting attendees, you know we have that provision in Nepal. So that budget was not sufficient actually."* (KII, Dhading)

FCHVs viewed free hydrocele surgery camps to be very effective but wished that it happened every year on a fixed schedule. They further explained that hydrocele patients usually do not open up and disclose to them about their condition, but when camp is organized, they tend to show up.

*"This program should be conducted on a regular basis. There should be a fixed routine for organizing camps so that people are more aware of it. Those who missed this year will be assured that they can come back again next year and get the treatment."* (FCHV, FGD, Kanchanpur)

**Hydrocele patients.** The study found that most of the patients did not have any idea about the program. In fact, we encountered only one patient who underwent surgery through the camp and he seemed content with the service. As for the others, the perception regarding the service was mixed with some mentioning that they did not want to take risks with surgery in government hospitals, which is why they went to private hospitals for surgery.

*"I heard that treatment is available, and it [camp] will arrive soon. I also went for an examination there. But some of my neighbors had undergone [hydrocele] surgery there previously, and it was not successful [post-surgery complications]. They suggested me to go to Kathmandu instead of having surgery here and not just worry about the cost as health is more important. So, we decided to go to Kathmandu."* (IDI, post-surgery, Dhading)

## Barriers for accessing free hydrocele surgery

**Hydrocele manifestation.** People only tend to seek and receive medical services, when their condition causes extreme pain and discomfort or has some risk of mortality. Generally, this trend is common, as confirmed in our study. Post-surgery respondents recalled their experience of having extreme pain and discomfort because of hydrocele in addition to feeling shame and embarrassment, and thus they decided to finally seek treatment. Meanwhile in the IDIs, most of the hydrocele patients who had not had surgery yet mentioned not feeling any pain and discomfort due to hydrocele and thus do not feel they need to have surgery.

*"I would have asked around [for treatment] if mine hurt. It doesn't hurt, so I didn't do anything, didn't ask anyone."* (IDI, pre-surgery, Kanchanpur)

Although infection can occur at an early age, manifestation of hydrocele happens usually at an older age. All the IDI respondents we identified in the study were above 40 years of age. Hence, old age could be another barrier to accessing the surgery.

*"I just didn't want to get treatment. You know I am old, what do I have to do, I just stay at home doing nothing. I felt there was no need for treatment. But it started to hurt slowly, so finally, I went."* (IDI, post-surgery, Dhading)

**Fear of surgery.** The study found that "fear of surgery" is one of the key barriers as well as preconceived notions among the respondents. It is presumable to say that the word "surgery" in itself instills fear and worry among most people. That fear of surgery coupled with "surgery of one's genitalia" acts like adding insult to the injury- with worry and confusion among the patients.

*"I got scared of the surgery. I thought it might not get bigger [even without surgery]. My friend's [hydrocele] got bigger within 3–4 months. People might avoid treatment because if*

*they go to the health facility, they will be asked for having surgery, so because of shame and fear, they might avoid it."* (IDI, pre-surgery, Kanchanpur)

It would not be an exaggeration to say that for any lay person, the very concept of surgery triggers certain fear in general. In hydrocele cases, in addition to the possible pain involved, the fear is further amplified because it involves the genitalia. Fear of surgery outcomes such as infertility or even death hindered people from getting the required treatment. More awareness and knowledge regarding surgery are essentially important to have more people elect to access the treatment offered.

**Mistrust in government services and accessibility.**   The study noticed that most of the patients had very low faith and trust in government services, which in turn could have affected the hydrocele surgery program as well. Patients mostly complained of a lack of qualified staff and inadequate number of doctors in government hospitals. Some respondents also recounted having bitter experiences in district hospitals such as negligence and rude behaviors by health workers while seeking other treatments and mentioned dissatisfaction with the services provided there.

*"The truth is they do not give any information for the poor. We take a loan and go for treatment, but they focus on taking our money before giving proper care first."* (IDI, wife of a patient, Kanchanpur)

*"There are many hydrocele patients in our district, to be honest. But I think due to fear, many patients didn't come this year due to last years' experience. There was lots of infection last year."* (FGD, FCHV, Dhading)

This issue of mistrust in their district healthcare could have further amplified the barriers to seeking care. For example, one stakeholder admitted that for most of the people in Kanchanpur, going to Seti zonal hospital in Dhangadi (another district) or even in India (as it is a bordering district to India) is more feasible than coming to Mahakali hospital. Dhading, on the other hand, is the closest district to the capital city of Kathmandu, and the patients from Dhading feel they would rather go to Kathmandu for better care.

*"Sadly, there is no service available here [Mahakali Zonal Hospital]. There are no capable doctors here, what to do. Many people go to India due to the lack of services here. If service was available here, people would not go to India."* (IDI, pre-surgery, Kanchanpur)

*"I admit that we have a severe lack of skilled manpower. On top of that Seti zonal hospital and India are very near from here. Due to that reason also, many people either prefer Dhangadhi or India, as service is trustworthy and easily accessible there."* (KII, Kanchanpur)

**Information dissemination and awareness.**   The limited and insufficient budget allocated for generating awareness and information dissemination concerning the surgery camp can be identified as one of the barriers since stakeholders from both Dhading and Kanchanpur informed that optimal information dissemination was not done, which in turn affected the patient turn-out rate. Similar to Dhading, out of 72 targeted, only 21 patients showed up during the hydrocele surgery camp last year.

*". . .use of many information dissemination media for awareness raising such as radio, TV, newspaper has not been used optimally as I have realized. Although we did dissemination*

*through pamphlets, posters, and FM radio, we couldn't do it effectively due to the ceiling in our budget."* (KII, Kanchanpur)

**Economic barrier.** Because surgery is considered crucial, many people believe that post-surgery requires a long resting time in order to gain back their strength and fitness. This would mean that they would have to avoid going to work and thus compromising their livelihood and income for an extended period. Since males are mostly the sole breadwinners of the family, for people living paycheck to paycheck, the decision to undergo surgery becomes a critical one.

*"After he gets his salary, we are planning to go [for surgery]. He can also rest after surgery for some time since driving will also be affected due to the rainy season coming soon. He has to rest for 1–2 months, at least. We have children to feed. What if the owner replaces another driver if he is absent for long time, you know?"* (IDI, wife of a patient, Kanchanpur)

Since the patients identified were mostly from city (district headquarters) areas, the study could not find accounts of geographical constraints from the patient side, but certainly, the patients expressed that transportation service and incentives for patients could serve as a motivation to access the healthcare services.

*". . .it is not easy for people living in rural areas to come to the headquarter for getting service. Even though the cost of surgery is free, the cost of transport coupled with days lost at work, not only of the patient but also one caretaker tagging along the patient accounts a lot for a poor person. What I mean is, if we could carry out mobile camps in communities closer to the settlement, then it would greatly increase the output."* (KII, Dhading)

In addition, district-level stakeholders and FHCVs alike were vocal about the importance of giving additional travel allowance for those patients who live far away from the district headquarter or exploring the possibility of conducting mobile surgery camps for those from hard to reach areas.

*"After surgery, if they could be provided with some minimal amount as transportation allowance, then I think it would motivate people to come and get treatment as it would cover some of their miscellaneous expenses."* (FGD, FCHV, Kanchanpur)

### Enablers for accessing free hydrocele surgery

The following themes were identified as enablers for accessing free hydrocele surgery as perceived by the stakeholders as well as hydrocele patients.

**Financial support and sufficient budget.** The budget for conducting the camp was a package cost of 6,000 rupees (around $55) per patient. The stakeholders agreed that the budget allocated for the hydrocele surgery camp was enough to bear the expenses that occurred, although the budget allocated for information dissemination prior to the camp was insufficient.

*"We didn't have any shortage of budget. In the first year, we were able to provide surgery to about 200 patients, and the budget was allocated accordingly. But due to lack of trained doctors, after the first year we have not been able to provide this service accordingly."* (KII, Kanchanpur)

**Flexibility of guideline.** According to the stakeholders, the program guideline is flexible in order to accommodate the need and context of place/health facility where the camp is planned. For example, in case of a lack of surgeons and infrastructures for conducting the camp, the hospital administration can coordinate with any private hospitals or agencies for necessary coordination, and hire a surgeon/consultant temporarily.

*"Well, the program guideline illustrates that if the [government] hospital does not have required human resource, Operation Theatre (OT) setup and other resources, they can coordinate with the private hospital without exceeding the budget ceiling."* (KII, Central level)

**Post-surgery experience.** Patients who already had surgery revealed that they feel extremely happy and comfortable after having surgery. Irrespective of where they had surgery, every one of them reported feeling confident and energetic after surgery and some even mentioned regretting not doing it any sooner.

*"It is all well now. It would have been better, had I done it earlier. I did it only after growing old. She [wife] says it's okay. It's alright now."* (IDI, post-surgery, Kanchanpur)

*"There has been a lot of changes. I can go anywhere. I can work any kind of job. I was ashamed and embarrassed before due to hydrocele. Now I am at peace. There is no tension now. I can walk anywhere with ease."* (IDI, post-surgery, Kanchanpur)

**Increased awareness and scale-up of the program in all remaining endemic districts.** People are more aware of and more informed about services and facilities than previous times as recounted by FCHVs during the FGDs. They further mentioned that people with the right information and knowledge are more willing to get the services compared to those who have no knowledge of hydrocele nor awareness about the services. Some hydrocele patients themselves were open about talking about their condition and shared that they are not ashamed of having hydrocele and are willing to get treated. In addition, stakeholders were optimistic about gradually scaling-up of program in all other endemic districts which means that more people can access the service, so it is hoped that the surgery turnout rate is going to be improved in the coming years.

*"Previously there were such misconceptions like; after surgery they might be unable to have kids. People used to be scared if their whole scrotum would be cut off. But I don't think people believe that anymore."* (FGD, FCHV, Kanchanpur)

## Discussion

### Knowledge and perception of hydrocele

The study found that the knowledge of the cause of hydrocele was very minimal among the respondents. Varied causal factors such as cold, trauma or injury, heavy physical work, and illicit sexual activity were attributed to the causation of hydrocele, while only one patient mentioned mosquito bite as the primary cause of hydrocele. But none of the respondents had any remote idea or mentioned about LF being associated with hydrocele. Similar mistaken beliefs have been reported in studies in countries like Kenya, and India dating back to the 1990's [18]. The fact that it still persists now is certainly worrisome and shows our intervention designs' inability to address local misperceptions and misunderstandings. Reassuringly, despite having no knowledge of the actual cause of hydrocele, people perceived hydrocele as a treatable

disease, and most of the patients had been to a medical practitioner for a check-up or for symptomatic pain relief at least once after having the hydrocele. Although societal level stigma and discrimination were not reported nor associated with hydrocele, however, it was clearly expressed that people with hydrocele felt like an outcast, and were noticed, talked and laughed about behind their backs. Hydrocele patients feel ashamed of themselves in participating in community gatherings and celebrations although the community is usually accepting and sympathizing [18]. The patients mentioned having low self-esteem, lack of confidence in themselves and suffering from extreme pain and discomfort as time passed.

All the patients reported to trying home remedies, with some even seeking faith healers before going to medical practitioners. A recent study in Nepal reports similar findings of LF patients visiting traditional faith healers and practicing home remedies before seeking medical help [15]. We can argue that it could be due to not knowing the disease etiology nor about the role of mosquitoes in the transmission of LF and attributing this to the prevalent beliefs in the society. Limited finances, lack of knowledge, and belief in traditional healing practices have been associated with low levels of health care utilization [19]. Since hydrocele manifests usually in the later age, with a gradual increase in size and pain, the patients' trend of seeking medical help was usually found to be much later after the first appearance of hydrocoele. Patients tended to seek medical treatment only when the disease seriously affected their livelihood [20]. According to the respondents, although pain could be seen as an apparent precursor for seeking treatment here, it is also very crucial to not rule out the inherent shame and self-discrimination the patients feel about seeking treatment. It is not presumptuous to draw an inference that as long as they did not feel pain, the patients would prefer to keep the disease private instead of making it known by seeking treatment. Many patients avoided accessing care for fear of being identified as LF patients and only contacted medical help once it hindered their work significantly [21]. Accordingly, it is imperative that the program providers and stakeholders understand and acknowledge that focusing only on meeting targets of surgery with centralized services might not be adequate.

## Knowledge and perception of free hydrocele surgery program

The camp-style approach is deemed suitable in the Nepalese context, which was also expressed by the stakeholders because not all hospitals are equipped with necessary infrastructures nor adequate human resources for conducting surgery as a mainstream service. Further, this approach is also recommended by the WHO since it does not require high-level facilities, although it should be performed by trained medical personnel [6]. Thomas *et al.* (2009) also reported similar success stories of mass surgery weeks in Nigeria in reaching a large number of hydrocele populations in a short amount of time [22]. Although the services are delivered through the government hospitals only, through networking, coordination, and outsourcing with private hospitals, effective service could be provided as per the guideline of the program, which was adopted by district hospitals as well. This flexibility of program guidelines in order to address the contextual needs was one of the enablers of the program. Adaptability (flexibility) and compatibility (contextual appropriateness) have been identified as two important characteristics of the implementation success. While adaptability refers to the programmatic aspect, compatibility refers to programs' capacity to address the provider's preferences, organizational needs as well as community needs [16]. In these terms, the study found that the LF Elimination Program in Nepal should revisit and reflect on its compatibility with community needs and preferences, to determine whether this 'one size fits all' approach could be tailored contextually based on community knowledge and perspectives because the study found no such endeavors from the program side. The role of FCHVs in creating awareness is

instrumental and their contribution to reaching community members is of paramount importance and widely recognized in Nepal [23]. Although FCHVs have been mobilized primarily for MDA and also identifying patients in MMDP, we found that their role could be further enhanced if they themselves were equipped with knowledge about LF and its morbidities. FCHVs confessed that sometimes people do not take them seriously since they cannot answer all of their queries, which creates a gap for people to learn about the disease in order for them to decide whether or not to access the services. In hydrocele cases, in particular, gender stigmatization further creates barrier among FCHVs and hydrocele patients. Given the commendable successes of the FCHV program in Nepal in safe motherhood program, with their instrumental role in community-level awareness raising, it might seem like a best option to tap onto the same resource for LF elimination program. However, it is crucial that the stakeholders and healthcare providers recognize the issue of gender stigmatization in this situation, before mobilizing FCHVs in communities, for addressing community-level barriers, as our study has shown. To be precise, male volunteers or health workers might prove to be a more suitable choice in this situation.

Sub-optimal information dissemination was identified as one of the barriers to accessing the free hydrocele surgery. The majority of the patients who were interviewed also mentioned having no information on the free surgery program and shared that they are willing to get surgery if it is provided free of cost. Little or no information on hydrocele surgery has been identified as a rectifiable weakness in the MMDP program [13]. A low level of knowledge among the target population acted as an obstacle among many control interventions [24]. Surgery was avoided in some cases because it was associated with perceived complications such as infertility, impotence, decreased physical strength, and even death [15]. This lack of knowledge of the free surgery camps, poor knowledge of hydrocele surgery procedures are further aggravated by the fact that generally, patients had some level of mistrust in government services. This reason for mistrust was supported and accepted by district stakeholder's view, who mentioned that easy accessibility to neighboring district hospital and more physical capacity to cater to the needs of the people, availability of competent and qualified doctors as compared to the constant vacant positions in the district hospital were cited as some of the reasons for the preference for other hospitals over the district hospitals both in Dhading and Kanchanpur. This is a key barrier for accessing services, since only those patients with the means and residing near the border districts could receive the surgery. Communication is a fundamental part of modern medicine [25], so it is an absolute need that stakeholders and health providers improve both communication and education side by side while providing the desired services. It is also important to not misrepresent the accessibility problem of the patients since the service is based in district hospitals only. One study done by the WHO on the accessibility of health services mentions that not providing user fees and transport costs can have a negative impact on accessing health services by the poor and vulnerable populations [12]. Travel cost, lost time from work, and indirect costs incurred while accessing distant government service were cited as barriers for accessing care by low-income participants [20, 21]. Mobile mass surgery camps have been proven to solve this discrepancy of accessibility and thus should be explored [26]. Therefore, scaling-up of these camps should be done by increased funding and allocating more human resources in order to meet the need of affected men living in remote and hard to reach areas [27]. Coordination between the local stakeholders, district health office and district hospitals should be strengthened by clearly defining the roles and responsibilities of the actors involved in order to explore these possibilities respective to their context and place.

Interviewing and engaging all focal personnel from the national level stakeholders to community volunteers and hydrocele patients is a key strength of this study. In addition, exploring the socio-economic barriers faced by hydrocele patients provides much-needed patient and

community level insights into the mostly centralized data driven policies. However, there are several limitations that should be considered in this study and addressed in future studies such as: (i) the small number of hydrocele patients was mostly limited to a relatively older-age group which might have limited the possibility of other emerging themes; (ii) the study setting was focused on semi-urban areas, surrounding the district headquarters which seriously limits generalization to the rural areas; (iii) the loss to follow-up on respondents who declined participating in the study; and (iv) there may have been some recall bias of hydrocele patients to recount their experiences of complications in the presence of the interviewer.

## Conclusions

The study explored provider, program and individual patient factors related to the hydrocele surgery coverage under the LF Elimination Program in Nepal. The study highlights the barriers faced by patients to access the free hydrocele surgery including socio-economic and cultural barriers as well as their limited knowledge and misperceptions of hydrocele, difficulties they face and other challenges. Furthermore, the study explores the limitations, opportunities of free hydrocele surgery program and potential considerations needed in order to improve the program output with input from national as well as district level stakeholders. Thus, the findings and recommendations could supplement efforts by the national LF Elimination Program to put more focus on hydrocele surgery in line with the national LF elimination target.

## Supporting information

**S1 File. Interview guidelines in English.**
(DOCX)

**S2 File. Interview guidelines in Nepali.**
(DOCX)

**S3 File. Categorised data sets.**
(DOCX)

**S1 Table. COREQ checklist.**
(DOCX)

## Acknowledgments

The study team would like to thank all the respondents who participated in this study for their time and information.

## Author Contributions

**Conceptualization:** Choden Lama Yonzon.

**Data curation:** Choden Lama Yonzon.

**Formal analysis:** Choden Lama Yonzon.

**Funding acquisition:** Choden Lama Yonzon.

**Investigation:** Choden Lama Yonzon.

**Methodology:** Choden Lama Yonzon.

**Project administration:** Choden Lama Yonzon, Sagun Paudel, Ashmita Ghimire.

**Resources:** Choden Lama Yonzon.

**Software:** Choden Lama Yonzon.

**Supervision:** Retna Siwi Padmawati, Raj Kumar Subedi, Elsa Herdiana Murhandarwati.

**Validation:** Choden Lama Yonzon, Sagun Paudel, Ashmita Ghimire.

**Visualization:** Choden Lama Yonzon.

**Writing – original draft:** Choden Lama Yonzon.

**Writing – review & editing:** Choden Lama Yonzon, Retna Siwi Padmawati, Raj Kumar Subedi, Sagun Paudel, Ashmita Ghimire, Elsa Herdiana Murhandarwati.

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
