## [Decision Letter · Decision Letter 0]

19 Aug 2020

PONE-D-20-21081

Exploring determinants of hydrocoele surgery coverage related to Lymphatic Filariasis in Kanchanpur and Dhading districts of Nepal: An implementation research

PLOS ONE

Dear Dr. Lama Yonzon,

Thank you for submitting your manuscript to PLOS ONE. After careful consideration, we feel that it has merit but does not fully meet PLOS ONE’s publication criteria as it currently stands. Therefore, we invite you to submit a revised version of the manuscript that addresses the points raised during the review process.

We look forward to receiving your revised manuscript.

Kind regards,

Yaobi Zhang, M.D., Ph.D.

Academic Editor

PLOS ONE

2. Please address the following:

- In your Methods section, please provide additional information about the demographic details of your participants. Please ensure you have provided sufficient details to replicate the analyses such as: a)  a description of any inclusion/exclusion criteria that were applied to participant inclusion in the analysis and b) a table of relevant demographic details.

- Please include additional information regarding the interview guide used in the study and ensure that you have provided sufficient details that others could replicate the analyses. For instance, if you developed a guide as part of this study and it is not under a copyright more restrictive than CC-BY, please include a copy, in both the original language and English, as Supporting Information. In addition, please include further details of the development and validation of this tool.

- Please ensure you have thoroughly discussed any potential limitations of this study within the Discussion section, including the potential introduction of biases during data collection and sampling.

- Please modify the title to ensure that it is meeting PLOS’ guidelines (https://journals.plos.org/plosone/s/submission-guidelines#loc-title). In particular, the title should be "specific, descriptive, concise, and comprehensible to readers outside the field" and in this case we have concerns that the title is long and contains errors of grammar. An alternative title suggestion is: "Exploring determinants of hydrocoele surgery coverage related to Lymphatic Filariasis in Nepal: An implementation research study".

Reviewers' comments:

Reviewer's Responses to Questions

**Comments to the Author**

1. Is the manuscript technically sound, and do the data support the conclusions?

Reviewer #1: Partly

Reviewer #2: Partly

2. Has the statistical analysis been performed appropriately and rigorously? 

Reviewer #1: I Don't Know

Reviewer #2: N/A

3. Have the authors made all data underlying the findings in their manuscript fully available?

Reviewer #1: Yes

Reviewer #2: No

4. Is the manuscript presented in an intelligible fashion and written in standard English?

Reviewer #1: No

Reviewer #2: Yes

5. Review Comments to the Author

Reviewer #1: The manuscript has the potential to make a critical contribution to published literature about barriers to access to hydrocele surgery. However, it needs some significant rewriting to link the findings to previous studies, clearly explain the methodology used, and clarify the data analysis conducted. The results also could benefit from inclusion of some quantitative statistics, e.g. of the 10 people interviewed, 7 knew the cause of hydrocele. Finally, the manuscript could benefit from an English-language editor; I did not include specific editorial suggestions in my comments below.

Specifically,

- The introduction section needs reworking and editing. For example, in some parts (e.g., lines 105-114) it reads like an advocacy document as opposed to a research article. As another example, referring to the LF Elimination Program consistently throughout would be useful; instead it is referred to at various times as the MOHP, EDCD, or the LF program.

- Lines 89-90. Please clarify what you mean by ‘scale up the MMDP component of the LF elimination program’ if MMDP services are already available in all endemic areas. This seems contradictory.

- Line 110. Please include references to past research on hydrocele surgery access here, as well as throughout the introduction. The discussion section has a few references, but more exist on hydrocele surgery barriers as well as access to essential surgery in general and would be useful to cite.

- The background section would benefit from including the elements on MMDP needed for WHO to validate the elimination of LF as a public health problem and how this research links to those elements/helps national LF elimination programs overcome barriers to elimination.

- Line 114. The study seemed to aimed to understand the barriers to surgical access as opposed to addressing them. Consider rewording. It also would be useful to clarify here or in the discussion what was to be done with this increased understanding – make policy recommendations to the MOHP? Share with global policymakers?

- Throughout the methods section, it would be useful to standardize presentation of details about in-depth interviews (IDI), key informant interviews (KII) and focus group discussions (FGD) and discuss in the same order in each sub-section. As written, it was slightly unclear who was included in each group, what the sampling methods were for each, what data analysis was done for each, etc. This would also hold true for presentation of the results – a consistent approach could help readers comprehend what was learned from each group of respondents.

- Why were the three family members included in the IDIs? To me, it is more interesting to see what responses comes from the patients vs all others and inclusion of the family members with the patients confuses the perspective of the patients.

- Line 138-140. Although mentioned that purposive sampling ensures diversity in terms of age, SES, urban/rural, the text states that the purposive selection was only based on in terms of highest number of hydrocele patients. Consider deleting 139-140. It is also unclear when snowball sampling was employed.

- Line 143: Related to this, in the results, clearly state number (%) of patients located and number of patients consenting to take part.

- Line 147: how many FCHVs took part in each FGD?

- Line 148: What was the purpose of the telephone inquiries prior to the interviews?

- Line 156: How was data triangulation done? By whom?

- Line 168: How was data saturation measured?

- Lines 176-177: Who transcribed the data? Who ensured data quality?

- Lines 179-184: This is very vague. Was any specific qualitative data analysis guidance followed to code and group the data? Did more than one person code the data? How was intercoder variance addressed?

- Lines 197-206: It could be helpful to organize this by IDI, KII, and FGD sociodemographic characteristics and include IDI median age (range), rate of refusal, # of years with hydrocele, occupation, # had surgery at govt camp vs private hospital; KII sex, age, # years in current position; and FGD age, # years in the position.

- Lines 278-280: As described, that is more budget than most LF programs allocate for hydrocele camps or routine hydrocele services. Might be useful to include some information about what activities the respondent felt were lacking, if that is available.

- The entire barriers section might flow better if it were organized by the steps in accessing services, e.g. first a patient needs to recognize they need surgery (hydrocele manifestation), second their fear of surgery needs to be overcome, then their mistrust of government services needs to be overcome, then they need to be aware of the camps, and finally economic barriers need to be overcome.

- The discussion section might flow better if knowledge and perception of hydrocele comes before the knowledge and perception of the free hydrocele surgery program.

- How were results shared with the MOHP and others responsible for improving services?

- What specific policy or other recommendations do the authors have to the MOHP and/or the LF elimination program to overcome these barriers?

- What recommendations do the authors have for other countries implementing similar programs?

Reviewer #2: The manuscript by Yonzon et al. aims to identify barriers and enablers of access to surgical services for LF-related hydrocoele in Nepal. Data for the study come from a series of qualitative surveys among hydrocele patients, family members, community health volunteers and other stakeholders from two high-burden districts of the country. This is an important study, as there is currently limited information in the literature on this topic, and the authors do a good job of placing this study within the wider context of the global lymphatic filariasis elimination effort and morbidity service provision. While the conclusions are supported by the data, the very small sample size of the hydrocele patients for the study (only 5 people who underwent hydrocele surgery and 4 hydrocele patients who had not) severely weakens the study. It does not invalidate the data obtained from the study participants and other study participant groups, but it does call into question the study design and representativeness of the core findings—particularly as these participants were spread over two districts.

Specific comments are provided below.

Major comments:

1. Sample size/study design: Why were so few hydrocele patients included in this study? How was snowball sampling employed if the numbers of participants in each survey type was so small?

2. The Results section is difficult to follow as the authors organize results thematically rather than by sample group (KII, FGD, IDI). The result is repeated switching between sample groups that disrupts the logical flow of the Results section. If possible, I recommend re-organizing results by sample group (KII, FGD, IDI) then highlighting specific themes within each group. If this is not feasible, then at least clarify in each paragraph which survey group is being described.

3. I recommend a table summarizing, by survey group, the socio-demographic characteristics and quantitative analysis of key questions (e.g. the proportion of respondents who know the cause of hydrocele; the proportion who know that it is transmitted by mosquito, etc.).

4. Did the study ask hydrocele patients about enablers to seeking surgery? I don’t see such data reported.

5. The Discussion should include a section acknowledge the limitations of the study—including, but not limited to the study’s small sample size, age bias in sample population, geographic bias, etc.

6. I appreciate that English likely is not the authors’ first language. However, the manuscript would benefit from editorial assistance to improve grammar and overall English.

Minor comments:

7. Introduction, 3rd paragraph: Suggest adding “in Nepal” to the sentence beginning on line 95 to clarify the paragraph’s contents relate to Nepal specifically.

8. Methods (line 130 and 133): please verify the numbers “1,71,304” and “3,36,067”.

9. Methods. I don’t see that “stakeholders” is defined. Who are they?

10. Methods: Research setting. Please add information on how many surgical camps have been conducted in the survey districts.

11. Please clarify which sample population is referred to in the first line of the results (line 197).

12. Please double check the author listing for reference #12 – it does not look correct.

6. PLOS authors have the option to publish the peer review history of their article (what does this mean?). If published, this will include your full peer review and any attached files.

Reviewer #1: No

Reviewer #2: No

---

## [Author Response · Author response to Decision Letter 0]

30 Sep 2020

ACADEMIC EDITOR COMMENTS: 

• Response: Thank you very much for the suggestions. We have followed the requirements.

2. In your Methods section, please provide additional information about the demographic details of your participants. Please ensure you have provided sufficient details to replicate the analyses such as: a) a description of any inclusion/exclusion criteria that were applied to participant inclusion in the analysis and b) a table of relevant demographic details.

• Response: Thank you for your suggestions. We agree with you and have incorporated this in the method section under the sub-head “Sampling and sample size” (page-9, Line208-210). A table depicting a sociodemographic profile has been added in the result section (page-12,13). 

3. Please include additional information regarding the interview guide used in the study and ensure that you have provided sufficient details that others could replicate the analyses. For instance, if you developed a guide as part of this study and it is not under a copyright more restrictive than CC-BY, please include a copy, in both the original language and English, as Supporting Information. In addition, please include further details of the development and validation of this tool.

• Response: Thank you for your insight. We have uploaded the interview guides as Supporting Information. Details are provided under sub-head “Data collection and research instruments” ( page-9, Line 218-224).

4. Please ensure you have thoroughly discussed any potential limitations of this study within the Discussion section, including the potential introduction of biases during data collection and sampling.

• Response: Thank you for raising an important issue. We have addressed this in our discussion section (page-31, final paragraph)

5. Please modify the title to ensure that it is meeting PLOS’ guidelines (https://journals.plos.org/plosone/s/submission-guidelines#loc-title). In particular, the title should be "specific, descriptive, concise, and comprehensible to readers outside the field" and in this case we have concerns that the title is long and contains errors of grammar. An alternative title suggestion is: "Exploring determinants of hydrocoele surgery coverage related to Lymphatic Filariasis in Nepal: An implementation research study".

• Response: Thank you for your suggestion. We have incorporated your suggestion in the revised manuscript. 

6. We note that you have indicated that data from this study are available upon request. PLOS only allows data to be available upon request if there are legal or ethical restrictions on sharing data publicly. For information on unacceptable data access restrictions, please see http://journals.plos.org/plosone/s/data-availability#loc-unacceptable-data-access-restrictions.

• Response: Thank you for your assessment. We have made de-identified and categorised data sets available as Supporting Information and other relevant data are within the paper. However, the raw transcripts generated cannot be made publicly available for ethical reasons. Public availability would compromise participants confidentiality as it contains personal information of the participants as well as references throughout most of the transcripts such as job title, location, which has the potential to identify the respondents. Our consent form explicitly states that the no information would be made available that could compromise their identity. Our institution does not have an established point of contact to field external request for access to raw data. Hence, additional relevant information can be made available on reasonable request to the corresponding author. We hope the presented data sets meets the requirement of the journal. 

• Response: Thank you for your suggestion. We have incorporated accordingly. 

REVIEWER 1 COMMENTS: 

1. The manuscript has the potential to make a critical contribution to published literature about barriers to access to hydrocele surgery. However, it needs some significant rewriting to link the findings to previous studies, clearly explain the methodology used, and clarify the data analysis conducted. The results also could benefit from inclusion of some quantitative statistics, e.g. of the 10 people interviewed, 7 knew the cause of hydrocele. Finally, the manuscript could benefit from an English-language editor; I did not include specific editorial suggestions in my comments below.

Specifically,

- The introduction section needs reworking and editing. For example, in some parts (e.g., lines 105-114) it reads like an advocacy document as opposed to a research article. As another example, referring to the LF Elimination Program consistently throughout would be useful; instead it is referred to at various times as the MOHP, EDCD, or the LF program.

• Response: Thank you very much for providing these insights. We agree with your assessment and have made significant changes in the introduction, methodology as well as result section. A table has been added to highlight the socio-demographic data quantitatively. The final copy of the manuscript has been edited by an English-language editor.

2. Lines 89-90. Please clarify what you mean by ‘scale up the MMDP component of the LF elimination program’ if MMDP services are already available in all endemic areas. This seems contradictory.

• Response: Thank you for your assessment. We agree that the sentence seemed rather contradictory. Thus we have explained further by giving a detailed description in page no. 5-6, line 112-124.

3. Line 110. Please include references to past research on hydrocele surgery access here, as well as throughout the introduction. The discussion section has a few references, but more exist on hydrocele surgery barriers as well as access to essential surgery in general and would be useful to cite.

• Response: Thank you for your suggestion. We have incorporated more references. 

4. The background section would benefit from including the elements on MMDP needed for WHO to validate the elimination of LF as a public health problem and how this research links to those elements/helps national LF elimination programs overcome barriers to elimination

• Response: Thank you for your assessment. We have incorporated your suggestion by giving a little background on WHO’s guideline on MMDP as well as national elimination program. It is reflected in page no. 4 (line 89-95) and page no. 5 (line 111-114).

5. Line 114. The study seemed to aimed to understand the barriers to surgical access as opposed to addressing them. Consider rewording. It also would be useful to clarify here or in the discussion what was to be done with this increased understanding – make policy recommendations to the MOHP? Share with global policymakers?

• Response: Thank you for your valid assessment. We have incorporated your suggestions. (page 7, line 151-154) and addressed more in the discussion section. 

6. Throughout the methods section, it would be useful to standardize presentation of details about in-depth interviews (IDI), key informant interviews (KII) and focus group discussions (FGD) and discuss in the same order in each sub-section. As written, it was slightly unclear who was included in each group, what the sampling methods were for each, what data analysis was done for each, etc. This would also hold true for presentation of the results – a consistent approach could help readers comprehend what was learned from each group of respondents.

• Response: We agree with your assessment and have incorporated changes throughout the methods and result section. 

7. Why were the three family members included in the IDIs? To me, it is more interesting to see what responses comes from the patients vs all others and inclusion of the family members with the patients confuses the perspective of the patients.

• Response: You have raised several interesting questions. In the beginning of the study, we also discussed whether including family members would be necessary or not. After discussion, we decided to include family members of hydrocoele patients who have not had undergone surgery yet, hoping that it would provide us some in-depth information on what the family or community thinks about hydrocoele and underlying reasons for not undergoing surgery (yet) in case hydrocoele patient would not disclose it openly.

8. Line 138-140. Although mentioned that purposive sampling ensures diversity in terms of age, SES, urban/rural, the text states that the purposive selection was only based on in terms of highest number of hydrocele patients. Consider deleting 139-140. It is also unclear when snowball sampling was employed.

• Response: Thank you for this insight. We have deleted the sentence. Some hydrocoele patients were identified with the help of those who participated in the interview. Hence both purposive and snowball sampling were used to identify IDI respondents.

9. Line 143: Related to this, in the results, clearly state number (%) of patients located and number of patients consenting to take part.

• Response: Thank you for the suggestion. We have incorporated in the result section.

10. Line 147: how many FCHVs took part in each FGD?

• Response: Seven FCHVs in Kanchanpur and 5 FCHVs in Dhading took part in FGDs.

11. Line 148: What was the purpose of the telephone inquiries prior to the interviews?

• Response: Telephone inquiries were done primarily to inquire about their availability as well as confirm their consent to participate. 

12. Line 156: How was data triangulation done? By whom?

• Response: Data triangulation was done by cross-checking data from different group of respondents, by the corresponding author. 

13. Line 168: How was data saturation measured?

• Response: Data saturation was insured by making sure all the questions and variables have been covered from all group of respondents and no new information was gathered. 

14. Lines 176-177: Who transcribed the data? Who ensured data quality?

• Response: Data was transcribed by principal interviewer (research assistant and corresponding author). Data quality was ensured by the principal interviewer and corresponding author. 

15. Lines 179-184: This is very vague. Was any specific qualitative data analysis guidance followed to code and group the data? Did more than one person code the data? How was intercoder variance addressed?

• Response: Thank you for your insights. Data coding and analysis were done manually by the corresponding author. 

16. Lines 197-206: It could be helpful to organize this by IDI, KII, and FGD sociodemographic characteristics and include IDI median age (range), rate of refusal, # of years with hydrocele, occupation, # had surgery at govt camp vs private hospital; KII sex, age, # years in current position; and FGD age, # years in the position.

• Response: Thank you for the suggestion. We have incorporated your feedback by adding a table depicting the socio-demographic profile of the respondents.

17. Lines 278-280: As described, that is more budget than most LF programs allocate for hydrocele camps or routine hydrocele services. Might be useful to include some information about what activities the respondent felt were lacking, if that is available.

• Response: We agree with your observation. In our revisions, we have attempted to address this more clearly and we think that the barrier section also highlights your suggestion, We hope that you agree (line 413 and 424).

18. The entire barriers section might flow better if it were organized by the steps in accessing services, e.g. first a patient needs to recognize they need surgery (hydrocele manifestation), second their fear of surgery needs to be overcome, then their mistrust of government services needs to be overcome, then they need to be aware of the camps, and finally economic barriers need to be overcome.

• Response: Thank you for your important insight. We have incorporated your suggestions accordingly in our revision. 

19. The discussion section might flow better if knowledge and perception of hydrocele comes before the knowledge and perception of the free hydrocele surgery program.

• Response: Thank you for your insightful observation. We have incorporated it accordingly.

20. How were results shared with the MOHP and others responsible for improving services?

• Response: The stakeholders were briefed about the preliminary findings during the time of KII. Detail findings were shared later on after completion of report writing online due to COVID-19 restrictions. 

21. What specific policy or other recommendations do the authors have to the MOHP and/or the LF elimination program to overcome these barriers?

• Response: We have reflected on this comment by incorporating arguments and recommendations related to our findings as well as other studies throughout the discussion section. 

22. What recommendations do the authors have for other countries implementing similar programs?

• Response: You have asked a very interesting question. We believe that contextual difference should be addressed by policy makers and stakeholders alike wherever this program is being implemented and tailored according to the ground reality instead of blanket approach. In fact, within countries also, approaches could or should be varied and flexible to cultural contexts. 

REVIEWER 2 COMMENTS:

The manuscript by Yonzon et al. aims to identify barriers and enablers of access to surgical services for LF-related hydrocoele in Nepal. Data for the study come from a series of qualitative surveys among hydrocele patients, family members, community health volunteers and other stakeholders from two high-burden districts of the country. This is an important study, as there is currently limited information in the literature on this topic, and the authors do a good job of placing this study within the wider context of the global lymphatic filariasis elimination effort and morbidity service provision. While the conclusions are supported by the data, the very small sample size of the hydrocele patients for the study (only 5 people who underwent hydrocele surgery and 4 hydrocele patients who had not) severely weakens the study. It does not invalidate the data obtained from the study participants and other study participant groups, but it does call into question the study design and representativeness of the core findings—particularly as these participants were spread over two districts.

Specific comments are provided below.

Major comments:

1. Sample size/study design: Why were so few hydrocele patients included in this study? How was snowball sampling employed if the numbers of participants in each survey type was so small?

• Response: Thank you for your important insights. We agree that the study employed a very small number of respondents and have thus acknowledged it as one of the limitations to our study. While collecting data, after making sure that all the variables were collected as per the interview guide, and no new information was obtained from the respondents, we felt it that it had reached saturation. Some hydrocoele patients were identified with the help of those who participated in the interview. Hence both purposive and snowball sampling were used to identify IDI respondents.

2. The Results section is difficult to follow as the authors organize results thematically rather than by sample group (KII, FGD, IDI). The result is repeated switching between sample groups that disrupts the logical flow of the Results section. If possible, I recommend re-organizing results by sample group (KII, FGD, IDI) then highlighting specific themes within each group. If this is not feasible, then at least clarify in each paragraph which survey group is being described.

• Response: Thank you for your suggestion. We believe this is a very valid assessment and have tried to incorporate this suggestion throughout result section making sure that the results are not repeated and the flow is maintained. 

3. I recommend a table summarizing, by survey group, the socio-demographic characteristics and quantitative analysis of key questions (e.g. the proportion of respondents who know the cause of hydrocele; the proportion who know that it is transmitted by mosquito, etc.).

• Response: Thank you for your suggestions. We have added a table highlighting the socio-demographic profile of the respondents (page 12, 13)

4. Did the study ask hydrocele patients about enablers to seeking surgery? I don’t see such data reported

• Response: Thank you very much for an important query. While defining enablers, we tried to focus on the factors which enabled the surgery (camp) to take place in general programmatic sense and thus identifying the barriers would highlight the issues that is needed to be addressed in addition to the already available conditions which made hydrocoele surgery possible in the first place. This is presuming that understanding and then addressing the identified barrier (patient side as well as programmatic side), would automatically enable the patient to access the service.

5. The Discussion should include a section acknowledge the limitations of the study—including, but not limited to the study’s small sample size, age bias in sample population, geographic bias, etc

• Response: Thank you for your important insight. We have incorporated your suggestion and made changes (page 31, line 718-728).

6. I appreciate that English likely is not the authors’ first language. However, the manuscript would benefit from editorial assistance to improve grammar and overall English.

• Response: We agree with you. Final manuscript copy has been edited by an English-language editor.

Minor comments:

7. Introduction, 3rd paragraph: Suggest adding “in Nepal” to the sentence beginning on line 95 to clarify the paragraph’s contents relate to Nepal specifically.

• Response: Thank you for your suggestion. We have revised the introduction section and have incorporated your suggestion. We hope you agree that these changes are better.

8. Methods (line 130 and 133): please verify the numbers “1,71,304” and “3,36,067”

• Response: Thank you for your important insight. We have made corrections.

9. Methods. I don’t see that “stakeholders” is defined. Who are they?

• Response: Thank you for important observation. We have clarified in the method section sub-head “research type and design” (page 7, line 167-168).

10. Methods: Research setting. Please add information on how many surgical camps have been conducted in the survey districts.

• Response: Thank you for your suggestion. We have incorporated accordingly (page 8, line 184-187)

11. Please clarify which sample population is referred to in the first line of the results (line 197)

• Response: Thank you for your insight. We have clarified that (page-11, line 262)

12. Please double check the author listing for reference #12 – it does not look correct.

• Response: We agree with you and re-checked the author listing. It might be due to the name of one of the author (deSilva), we believe it is the correct listing as the author’s name is Nilanthi R. deSilva.

---

## [Decision Letter · Decision Letter 1]

11 Nov 2020

PONE-D-20-21081R1

Exploring determinants of hydrocele surgery coverage related to Lymphatic Filariasis in Nepal: An implementation research study

PLOS ONE

Dear Dr. Lama Yonzon,

Thank you for submitting your manuscript to PLOS ONE. I apologize for the delayed review process and response. This was due to non-response from a reviewer who agreed to review. I have now looked at it myself and I agree with the reviewer on the minor points he raised. After careful consideration, we feel that it has merit but does not fully meet PLOS ONE’s publication criteria as it currently stands. Therefore, we invite you to submit a revised version of the manuscript that addresses the points raised during the review process.

We look forward to receiving your revised manuscript.

Kind regards,

Yaobi Zhang, M.D., Ph.D.

Academic Editor

PLOS ONE

Reviewers' comments:

Reviewer's Responses to Questions

**Comments to the Author**

1. If the authors have adequately addressed your comments raised in a previous round of review and you feel that this manuscript is now acceptable for publication, you may indicate that here to bypass the “Comments to the Author” section, enter your conflict of interest statement in the “Confidential to Editor” section, and submit your "Accept" recommendation.

Reviewer #2: (No Response)

2. Is the manuscript technically sound, and do the data support the conclusions?

Reviewer #2: Yes

3. Has the statistical analysis been performed appropriately and rigorously? 

Reviewer #2: N/A

4. Have the authors made all data underlying the findings in their manuscript fully available?

Reviewer #2: Yes

5. Is the manuscript presented in an intelligible fashion and written in standard English?

Reviewer #2: Yes

6. Review Comments to the Author

Reviewer #2: The authors have done an adequate job of addressing the majority of issues highlighted in the previous version, including all the major concerns raised. A few minor issues remain unaddressed, as well as several novel issues introduced by the revisions.

Minor comments (line numbers refer to the clean version of the revised manuscript):

1. Abstract, line 36: it would be useful to state the type of persons interviewed (e.g. national and district LF focal persons), as “stakeholders” is not defined in the abstract and can refer to a wide variety of individuals both internal and external to the program.

2. Introduction, line 107. The WHO process for LF elimination as a public health problem is “validation”, not “certification”.

3. Introduction, line 107-109: Given that it is now late 2020, the sentence beginning, “Nepal is also gearing-up to achieve LF elimination by 2020; ….” does not seem realistic. Stating MDA has reached 100% coverage is also not relevant toward this goal. What is the percentage districts that have stopped MDA? Completed TAS-3?

4. Methods (lines 178 and 182): please correct the format for numbers given in “X,XX,XXX” format.

5. Methods (line 213-214): “Written and informed consents”. Do the authors mean “Written informed consent”? All consent should be informed consent.

6. Discussion (line 684): The statement that “Stakeholders and healthcare providers should consider this issue in order to use the already established network of FCHV….” does not comport with the finding that “gender stigmatization further creates barrier among FCHVs and hydrocele patients”. Suggest revising.

7. Discussion (line 692): suggest adding “perceived”: “Surgery was avoided in some cases because it was associated with perceived complications such as…..”

8. The incorrect reference was #12 in the original manuscript; #21 in the revised manuscript. The correct citation is Thomas et al., not Jindau et al.

7. PLOS authors have the option to publish the peer review history of their article (what does this mean?). If published, this will include your full peer review and any attached files.

Reviewer #2: No

---

## [Author Response · Author response to Decision Letter 1]

13 Dec 2020

Re: Resubmission of manuscript, “Exploring determinants of hydrocele surgery coverage related to Lymphatic Filariasis in Nepal: An implementation research study”, PONE-D-20-21081R1

Yaobi Zhang, M.D., Ph.D.

Academic Editor

PLOS ONE

Dear Dr. Zhang,

Thank you once again for inviting us to submit a revised draft of our manuscript titled, “Exploring determinants of hydrocele surgery coverage related to Lymphatic Filariasis in Nepal: An implementation research study” to PLOS ONE. We really appreciate the time and effort you and each of the reviewers have dedicated by providing insightful feedback to further improve our paper. We have incorporated changes that reflect the suggestions you have graciously provided. We hope that the responses we have provided, and the changes made satisfactorily address all the concerns you and the reviewers have noted. 

 To facilitate your review of our revisions, point-by-point responses to the comments are provided below. 

REVIEWERS’ COMMENTS: 

Reviewer #2: The authors have done an adequate job of addressing the majority of issues highlighted in the previous version, including all the major concerns raised. A few minor issues remain unaddressed, as well as several novel issues introduced by the revisions.

Minor comments (line numbers refer to the clean version of the revised manuscript):

1. Abstract, line 36: it would be useful to state the type of persons interviewed (e.g. national and district LF focal persons), as “stakeholders” is not defined in the abstract and can refer to a wide variety of individuals both internal and external to the program.

• Response: Thank you very much for your insights. We agree with your assessment and have made changes accordingly. 

2. Introduction, line 107. The WHO process for LF elimination as a public health problem is “validation”, not “certification”.

• Response: Thank you for your assessment. We have made the correction. 

3. Introduction, line 107-109: Given that it is now late 2020, the sentence beginning, “Nepal is also gearing-up to achieve LF elimination by 2020; ….” does not seem realistic. Stating MDA has reached 100% coverage is also not relevant toward this goal. What is the percentage districts that have stopped MDA? Completed TAS-3?

• Response: Thank you for your valid assessment. We agree with you and have thus revised the section which reflects recent data. 

4. Methods (lines 178 and 182): please correct the format for numbers given in “X,XX,XXX” format.

• Response: Thank you for your assessment. We have made the corrections accordingly. 

5. Methods (line 213-214): “Written and informed consents”. Do the authors mean “Written informed consent”? All consent should be informed consent.

• Response: Thank you for your valid assessment. We have corrected the sentence. 

6. Discussion (line 684): The statement that “Stakeholders and healthcare providers should consider this issue in order to use the already established network of FCHV….” does not comport with the finding that “gender stigmatization further creates barrier among FCHVs and hydrocele patients”. Suggest revising.

• Response: We agree with your assessment and have incorporated changes as reflected in page 29-30, line 685-692. 

7. Discussion (line 692): suggest adding “perceived”: “Surgery was avoided in some cases because it was associated with perceived complications such as…..”

• Response: Thank you for your valid insight. We agree with you and have made the correction accordingly. 

8. The incorrect reference was #12 in the original manuscript; #21 in the revised manuscript. The correct citation is Thomas et al., not Jindau et al.

• Response: Thank you for this important insight. We agree with you and have thus made corrections.

Thank you once again for giving us the opportunity to strengthen our manuscript with your valuable comments and suggestions. We have worked hard to incorporate your feedback and hope that these revisions persuade you to accept our submission. 

Sincerely,

Choden Lama Yonzon

Corresponding Author

Universitas Gadjah Mada

Yogyakarta, Indonesia

yonzon.chhoden.cy@gmail.com

---

## [Editor Report · Decision Letter 2]

15 Dec 2020

Exploring determinants of hydrocele surgery coverage related to Lymphatic Filariasis in Nepal: An implementation research study

PONE-D-20-21081R2

Dear Dr. Lama Yonzon,

We’re pleased to inform you that your manuscript has been judged scientifically suitable for publication and will be formally accepted for publication once it meets all outstanding technical requirements.

Kind regards,

Yaobi Zhang, M.D., Ph.D.

Academic Editor

PLOS ONE

---

## [Editor Report · Acceptance letter]

18 Feb 2021

PONE-D-20-21081R2 

Exploring determinants of hydrocele surgery coverage related to Lymphatic Filariasis in Nepal: An implementation research study 

Dear Dr. Lama Yonzon:

I'm pleased to inform you that your manuscript has been deemed suitable for publication in PLOS ONE. Congratulations! Your manuscript is now with our production department. 

Kind regards, 

on behalf of

Dr. Yaobi Zhang 

Academic Editor

PLOS ONE